# A Dutch Perspective on Two Health Related Issues Regarding Children and Adolescents with Intellectual Disabilities

**DOI:** 10.3390/ijerph191811698

**Published:** 2022-09-16

**Authors:** Xavier Moonen, Dederieke Festen, Esther Bakker-van Gijsel, Jessica Vervoort-Schel

**Affiliations:** 1Ben Sajet Center, Zwanenburgwal 206, 1011 JH Amsterdam, The Netherlands; 2Department of Child Development and Education, University of Amsterdam, 1018 WS Amsterdam, The Netherlands; 3Koraal Center of Expertise, De Hondsberg, Hondsberg 5, 5062 JT Oisterwijk, The Netherlands; 4Erasmus MC, University Medical Center Rotterdam, Postbus 2040, 3000 CA Rotterdam, The Netherlands; 5Radboud University Medical Center, Postbus 9101, 6500 HB Nijmegen, The Netherlands

**Keywords:** ID-physician, pediatric care, adverse childhood experiences, intellectual disabilities, children, adolescents

## Abstract

In this opinion article, we want to inspire readers by highlighting recent Dutch developments about two important health related issues regarding the quality of life of children and adolescents with intellectual disabilities. Firstly we focus on the prevention, treatment and reduction of (disability-related) somatic and psychological problems by specialized physicians for people with intellectual disabilities. Secondly, we emphasize the importance of the prevention of adverse childhood experiences and the promotion of protective and compensatory experiences. Subsequently, we stress the need for trauma informed care to support children and adolescents with intellectual disabilities who encounter adverse events. A specialized and multidisciplinary approach is advised as is the need for promoting healthy (family) relations with a focus on (co)regulation and connection as a basis for recovery.

## 1. Introduction

### 1.1. Intellectual Disabilities

According to the American Association on Intellectual and Developmental Disabilities, intellectual disabilities (IDs) are characterized by significant limitations in both intellectual and adaptive functioning. Intellectual functioning relates to the general mental capacity and involves learning, reasoning, and problem-solving. To assess intellectual functioning, an IQ test can be the appropriate instrument. Generally an IQ test score of up to 70 or 75 is an indicator of IDs. Adaptive functioning refers to the collection of conceptual, social and practical skills people learn and apply in their everyday lives. An impairment in adaptive functioning is considered significant when someone cannot behave as is expected to be adequate, based on age and culture [1].

### 1.2. The Dutch Approach to the Concept of Mild Intellectual Disabilities

In The Netherlands, the term ‘mild intellectual disabilities’ also refers to people with a total IQ score of up to 85 (borderline intelligence) who have significant problems in adaptive functioning. This is in line with the last *Diagnostic and Statistical Manual of Mental Disorders* criteria [2]. Approximately 440,000 people are considered to have an intellectual disability [3]. This concerns approximately 2.5% of the Dutch population. When people with borderline intelligence, who have severe deficits in adaptive functioning, are considered to have a mild intellectual disability as well, about 1.1 million people are involved, accounting for about 6.8% of the Dutch population [4]. The etiology of IDs is diverse and includes, for instance, genetic abnormalities, fetal exposure to alcohol/drugs, infectious diseases during pregnancy and traumatic brain injury during childbirth.

### 1.3. Two Health Related Issues Regarding People with IDs

In this contribution, we want to address two health related issues that are extremely important for the quality of life of children and adolescents with IDs. In Section 2, we discuss the work of specialized ID physicians for people with IDs. Specialized ID physicians provide high-quality evidence-based medical care to children, adolescents and adults with IDs. This care is aimed at preventing, treating and reducing (disability-related) somatic and psychological problems, in collaboration with other professionals, to ensure an optimal quality of life within the given limitations. We close this section with an illustration of how a specialized ID physician in The Netherlands can add value to the care of an adolescent with IDs. This case is presented with the permission of the legal representatives. In Section 3, we discuss the concepts of adverse childhood experiences (ACEs), protective and compensatory experiences and resilience in children and adolescents with IDs. Furthermore, we address (the treatment of) trauma and the need for trauma informed care. In Section 4, we present some closing remarks.

## 2. The Specialized ID Physician for People with Intellectual Disabilities in the Care of Children and Adolescents in The Netherlands

### 2.1. A Unique Dutch Medical Specialism: The Specialized ID Physician

Compared to the general population, individuals with IDs and their families have a higher risk of poorer overall health quality, behavioral and psychiatric disorders and serious physical health conditions [5,6]. They face numerous health disparities throughout their lives [7,8]. Since the year 2000, the medical specialism known as intellectual disability medicine has been acknowledged by the Dutch Minister of Health. Physicians who complete the 3-year vocational post graduate training are registered as specialized physicians for people with IDs. This medical specialism is—to our knowledge—only officially recognized in The Netherlands. The specialized ID physician provides (medical) care to people with IDs of all ages, and pays attention to the life-course perspective and the different phases within it. The specialized ID physician is usually affiliated with a care organization for people with IDs. In addition, the specialized ID physician is available through outpatient clinics for children and adults with intellectual disabilities who live with their parents or who live (under supervision) independently [9].

### 2.2. Specialized ID Physicians Collaborating with Other Professionals

Children with IDs may have multiple morbidities. Therefore, an extensive care network is often needed, depending on the pathology. In children with somatic problems, the pediatrician is usually the coordinating physician until age 16–18. The child psychiatrist, pediatric neurologist, rehabilitation specialist or the specialized ID physician may be involved in close collaboration either for a short period of time or long-term depending on the specific questions at hand. The multidisciplinary team may also include a behavioral therapist, a psychologist, an orthopedagogy expert, a speech and language therapist, a physiotherapist, an occupational therapist, etc. ID physicians are involved in complex and specific ID related issues such as: (1) complex and multifactorial symptoms, for example problem behavior and sleep problems; (2) special diagnostics and treatment methods; (3) questions related to specific phases of life, for example contraception or the desire to have children.

### 2.3. A Guideline for a Safe Transition into Adulthood

Children with IDs who are being treated by a pediatrician or another medical specialist usually make the switch from pediatric care to adult care between the ages 16 and 18. Transition in care involves preparation, the actual transition and aftercare and may be difficult, because often a long-term relationship with the pediatric team has been established. In addition, important changes regarding sexuality and psychosocial aspects, especially the transition to independent functioning, occur during adolescence and young adulthood, and require a different type of care, in which the pediatrician is (mostly) untrained.

Ensuring a safe and effective transition is a key issue in assuring the quality of care for adolescents with IDs. In The Netherlands, a guideline has been developed for the transition of care for adolescents with IDs by the associations for pediatricians, pediatric neurologists, pediatric rehabilitation specialists and specialized ID physicians [10]. In this guideline, the specialized ID physician has an important coordinating role for the adolescents and (young) adults with IDs (although also other medical doctors can fulfil this coordinating role). In addition to the specialized ID physician, a so-called transition coordinator plays an important role; drawing up an individual transition plan, and acting as the contact person for the adolescent and their caregivers/parents.

In the guideline requirements for the transition of care for adolescents with IDs, the following are discussed: (1) Raise the subject of transition in a timely manner (at least from the age of 14); (2) Provide a good written transfer including history; (3) The adult specialist should be a generalist (such as a specialized ID physician), although medical specialists, e.g., a cardiologist, may need to be involved as well; (4) Organize at least one joint multidisciplinary team consultation; (5) Appoint a transition coordinator who arranges practical matters; (6) Focus on more than just medical problems.

### 2.4. An Illustration of How a Specialized ID Physician Adds Value to the Care of an Adolescent with IDs

Q is a 15 year old girl. She walks independently, makes noises and knows how to swipe to another page on her iPad. Her greatest joy is to show her mother, on her iPad, what she wants to eat. The psychological report shows that she is diagnosed with severe IDs. The child neurologist in the academic university hospital is her principal, leading, clinician. Some five years ago, Q did not want to eat anymore. One week before this symptom started, her whole family had a stomach flu, with diarrhea and vomiting. As a result, Q lost weight, so the child neurologist decided to admit her to the hospital. Many diagnostic tests were run; an MRI scan, an EEG, a swallow picture, whole exome sequencing, blood tests, etc. None of these tests gave an explanation for her behavior. During the admission, a multidisciplinary consultation took place including all of the professionals involved in the multidisciplinary team. Q’s mother, who was familiar with the specialized ID physician, asked the team if the specialized ID physician could join the multidisciplinary meeting. During the hospital admission, Q did not want to drink, which induced constipation. On the specialized ID physician’s advice, a low dose of Locust bean gum was used to thicken her drinks, to enable her to eat the liquid. The specialized ID physician also advised the use of the Bristol Stool Chart to evaluate Q’s constipation. It turned out that Q had recently started using Levetiracetam to treat seizures. According to the experience of the specialized ID physician, some children with IDs experience behavioral side effects during the use of Levetiracetam. The Levetiracetam was gradually reduced and replaced with Oxcarbazepine. Another hypothesis was that Q had experienced the severe vomiting during the stomach flu as very traumatic. The specialized ID physician advised to consult a psychologist specialized in helping people with IDs, to conduct ye movement desensitization and reprocessing (EMDR). It turned out that the specialized ID physician was able to discuss all of the different topics with the different members involved in the multidisciplinary team. All of these different, but related problems are the domain of the specialized ID physician who is an expert in these kinds of complex cases. Together with the parents and the multidisciplinary team, it was decided that the specialized ID physician would stay involved and continued to be involved after Q reached the age of 18.

## 3. Adverse Childhood Experiences, Protective and Compensatory Experiences, Trauma, Resilience and Trauma Informed Care

### 3.1. Adverse Childhood Experiences

From the illustration above, it becomes clear that this child with IDs experienced events that negatively influenced her quality of life. In fact, experiences in childhood, both positive and negative, are strongly linked with health throughout life [11]. In the original adverse childhood experiences (ACEs) framework presented by Felitti and colleagues, ACEs were defined as experiences of abuse, neglect or household dysfunction that significantly contribute to negative health outcomes throughout life [12]. In 2014, the concept was clarified and defined as “childhood events, varying in severity and often chronic, occurring within a child’s family or social environment that cause harm or distress, thereby disrupting the child’s physical and psychological health and development” [13] and in 2020, Alhowaymel and colleagues added: “ACEs are influenced by globally diverse cultural, social, environmental, and economic factors that affect individuals’ health worldwide” (p. 22) [14]. Nearly two-thirds of school-aged youth experience a significant number of adverse advents, regardless of where they live in the world [15]. In a recent Dutch study, it was found that in a specific population, 81.7% of the children with IDs and 92.3% of the children with borderline intellectual functioning experienced at least one ACE, and about 20% of the children with moderate and mild intellectual disabilities experienced four ACEs or more. [16].

### 3.2. The Negative Effects of ACEs on (Mental) Health Outcomes

Over the past two decades in multiple studies, an association was found between ACEs and poor health outcomes, both mental and physical [17,18,19]. ACEs are associated with major health and financial costs across European countries [20] and experts consider them the leading cause of mental disorders and physical health damage worldwide [21,22]. Toxic stress caused by ACEs during childhood can have detrimental effects on the immune, endocrine and nervous systems, and can lead to epigenetic changes in the DNA structure. The developmental trajectories of children can be negatively impacted by the timing of adverse advents, their intensity and the consequences of the subsequent toxic stress [23]. In addition, the interaction with the family environment, the type of adversity and the pre-existing characteristics of the child also influence health outcomes. More so than children with average intelligence or average adaptive functioning, children with IDs are at risk of experiencing ACEs, which can cause severe behavior and health problems during their lifetime [24]. Due to their cognitive challenges, limited emotional regulation skills and increased dependence on others, children with IDs are hypothesized to be more susceptible to the harmful effects of ACEs [25]. The existing social inequalities in the health care of children with IDs may also be exacerbated by the unaddressed and untreated implications of ACEs. In The Netherlands, attention to research on ACEs is increasing, see for example [26,27,28,29,30,31,32].

### 3.3. Protective and Compensatory Experiences (PCEs) and the Resilience in People with Intellectual Disabilities

In recent years, scientific attention to positive childhood experiences is increasing as well, because of their preventive and mitigating effects on ACEs, thereby contributing to a healthy development and well-being in life [18,33]. International research regarding positive experiences in childhood, also referred to in research as protective and compensatory experiences (PCEs) [34,35] or advantageous childhood experiences [36], counter-ACEs [36] or benevolent childhood experiences [37] is growing. PCEs are experiences that promote resilience in the face of adversity [37]. Resilience is a dynamic, systemic process, based both on internal and external sources [38]. Relationships and resources form the basis for the PACE framework proposed by Hays-Grudo and Morris [35] that includes, for example, unconditional love and living in a home that is clean, safe, and with sufficient food. Unfortunately up until now, there has been little focus on the role of ACEs in research with regards to children and adults with IDs [39] and the preventive and buffering role of PCEs. The findings from a systematic review by Crompton et al. suggest that this is a highly neglected area of research [25].

### 3.4. Causes of Possible Trauma in People with IDs

Exposure to various potentially traumatic events can sometimes be directly related to disabilities, for example, through a medical condition, as presented in the illustration. ACEs are a second possible cause. However, trauma in people with IDs can also be related to the early separation from caregivers and institutionalization, or the increased vulnerability of people with IDs to the wrong intentions of others [40]. The dependence on care from others can increase the risk of care-giver-initiated violence [41]. From an intervention perspective, McNally and colleagues (p. 928) argue that recovery from traumatic experiences by people with IDs may be impeded by the “limitations in their ability to describe their experiences, locate and describe the associated emotions and their challenges to finding agency over their own lives” [24].

### 3.5. How to Deal with Trauma in People with IDs

Communication problems and the risk of diagnostic overshadowing (i.e., the attribution of a person’s symptoms to the ID when such symptoms actually suggest a pathological condition), as well as the lack of well-trained therapists [40] may contribute to under-reporting, under-diagnosing and the under-treatment of trauma-related disorders in people with IDs [42]. In Byrne’s systematic review [40], the preliminary findings suggest that cognitive behavior therapy (CBT) and EMDR, as presented in the illustration, are both viable and acceptable treatment options for children and adults with various levels of IDs. However, further high-quality research is needed regarding the specific treatment in people with IDs, as well as research on treatment effectiveness.

### 3.6. The Importance of Focusing on Families and the Living Environment as Well

Recovery and healing do not only take place in individual therapy, but mostly in the living environment of the child. A context with maximized psychological and physical safety, based on (co)regulation and connection, provides the base for recovery [43]. Insights from studies on neuroplasticity, epigenetics, and resilience may help to explain the wide variation in the effect of ACEs and traumatic experiences. These insights stress the importance of nurturing relationships and environments to increase the ability to buffer negative effects, build resilience, and heal [16]. As is shown in the illustration, it is important to focus on family functioning and the social environment when intervening in the development of children and adolescents, especially those with IDs, because of their dependence on caregivers and the social environment for their opportunities in development and health. The keys to a healthy brain, as well as to positive social, emotional, cognitive and physical development and well-being throughout childhood and adulthood are attuned, positive, safe, stable and nurturing relationships (SSNRs) and a healthy attachment between children and their primary caregivers [44]. Dysfunctional behavior in families can lead to ACEs and a lack of PCEs in children’s lives, which can obstruct SSNRs and can lead to problems in their later development and their chances of preparing for adulthood, leading to a cycle of intergenerational transmission of health damage [23]. Raising a child with special health needs, such as children with IDs, can also contribute to family dysfunction because of the increased stress it can cause [5,23]. In a Dutch study, the importance was addressed of social support for parents with IDs in regulating high parenting stress due to child behavior problems [45].

### 3.7. Trauma Informed Care

Toxic stress and unrecognized and untreated trauma symptoms may affect a person’s willingness to seek professional help, influences the working alliance in a negative way and may contribute to premature termination of help or failure to achieve desired results [46]. The first insights in The Netherlands regarding the prevalence of ACEs in children with IDs [32] prompted a call for action. PCEs and ACEs can enable and obstruct SSNRs, and supporting environments for children require adequate prevention and intervention, not only at the individual or family level, but also at the organizational level [24]. Such a developmentally supportive and restorative climate for all individuals involved, can be created by implementing trauma informed care (TIC). TIC is a system-level intervention that provides an organizational structure and treatment framework, and influences the culture of an organization [47]. TIC can be seen as a “system development model that moves away from the traditional diagnosis model of “what is wrong with you”, towards a story-based approach of “what happened to you” [47]. TIC has four key elements, referred to as the four Rs: Realizing the widespread impact of trauma, Recognizing how trauma may affect individual clients, staff or others in the program, Responding by applying knowledge about trauma into practice and Resisting retraumatization [48].

### 3.8. Trauma Informed Care in The Netherlands

In The Netherlands, trauma informed care is still a relatively new paradigm in the care for children and adolescents with IDs. There is growing evidence supporting its effectiveness in promoting resilience and preventing retraumatization in the general population [24,49] and it is a promising approach in the education and care for children and adolescents with IDs [50]. A better understanding of the life course associations between ACEs and IDs and the impact of exposure to multiple types of childhood adversities on disabilities and health is needed, to inform research and services focused on this vulnerable population [25]. Research on the implementation of TIC in youth care, education and long-term care for children and adults with cognitive and adaptive disabilities has recently started in The Netherlands. The ARTIC scale is used to measure professional and paraprofessional attitudes favorable or unfavorable toward TIC [51]. This scale has been translated into Dutch in accordance with the WHO guidelines and is currently being validated in The Netherlands.

## 4. Closing Remarks

Raising a child with IDs can be a challenging task for any parent. However, when families face adverse health and/or socioeconomic conditions, dysfunctional parenting patterns can emerge. The ID physician can, often in collaboration with a multidisciplinary team, play an important role in the support of the child with IDs and their family. It is important to counter adverse conditions that can negatively influence the development of a child with IDs, but professionals should also look for and value the importance of protective and compensatory child experiences and promote resilience. Professionals should not only focus on the behavioral, physical and mental health issues of the child with IDs, since a focus on the children’s family context and related social determinants of health is necessary too. For this, a trauma informed environment is very important. As shown in this contribution, more research is needed to support these developments.

## Data Availability

Not applicable.

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
