# Peer review of "A Dutch Perspective on Two Health Related Issues Regarding Children and Adolescents with Intellectual Disabilities"

_ijerph, 2022, doi:10.3390/ijerph191811698_

Round 1
Reviewer 1 Report
The authors present a manuscript describing a collaboration that examined two important issues related to life quality for children who have intellectual disabilities in the Netherlands. Overall, the manuscript is well conceived and well written. The authors highlight unique areas that have been accomplished in the Netherlands such as the establishment of a specialized ID physician and guidelines for transition to adulthood for this population. They also discuss the impact of Adverse Childhood experiences (ACES) and protective and compensatory experiences for this population. The article will make a great contribution to the field. Although editing for English is needed, no other areas are indicated.
Author Response
We thank the reviewer for the compliments. After revising the article English editing was performed.
Reviewer 2 Report
Review of
Some current issues regarding children and adolescents with intellectual disabilities in the Netherlands
I found the manuscript interesting and useful, it helped me better understand how people with intellectual disability are provided care in the Netherlands. That being said, for an international reader, some comparisons with other countries would really make the manuscript more convincing as now it reads more as an opinion piece, not a scientific article.
Here are some comments that I hope will help the authors improve their manuscript:
1. Consider shortening the first keyword or splitting into several keywords.
2. I am not sure whether the first paragraph of the introduction serves any real purpose or is relevant to the rest of the paper. Perhaps illustrate the current state of the arts in this particular field in the Netherlands in fewer words?
3. Please try and make your paragraphs shorter and try to have only one train of thought throughout one paragraphs. This will help the readers a lot.
4. Please make sure that after any factual claim you provide a citation supporting that claim, even if the claim may seem trivial. This will direct the readers to further literature to explore on the topic. It will also make your manuscript more substantial.
5. The manuscript would benefit from thorough language editing.
6. I am not sure of the purpose of providing a description of a case in chapter 2.1. Also, do such case descriptions not require approval from the ethics board and informed consent? Please look into this, this is important.
7. I strongly advise the authors to add chapters to the manuscript that would thoroughly describe and compare how other countries deal with individuals with intellectual disability and how care is provided to them.
8. As the saying goes, “when the only tool you have is a hammer – everything looks like a nail.” With that in mind, a substantial part of the manuscript should be devoted in convincing not only that a particular issues exits in the way people with intellectual disability are provided care in the Netherlands, but that devoting additional effort would be within the capability of the healthcare system. In essence, everyone has important issues they want to fight for, but in the grand scheme of things, from an impartial perspective, how practical would it be to direct the finite funds of the healthcare system toward the issues discussed in the paper?
Overall, there is still much work to be done to make the manuscript publishable. Mainly, the manuscript needs to be more relevant to the international reader and needs much more grounding in current literature. Hopefully, the authors will not be discouraged by my comments and will try to address the outlined issues, but that would mean almost rewriting the whole manuscript.
I wish the authors the best of luck in their ongoing and future research.
Author Response
Reply to reviewer 2
We thank the reviewer for the thoughtful comments. Because of this we thoroughly rewrote the manuscript. Since our purpose was not to present a research article but to provide an opinionating and inspiring contribution we clarified this in the abstract.
- We altered the first keyword
- The whole manuscript was thoroughly rewritten as was the introduction paragraph
- The whole manuscript was thoroughly rewritten trying to have only one train of thought throughout every paragraph
- Citation and notes we rewrote according to this remark
- Language editing was performed
- The purpose of the case was clarified in this rewritten version and is highlighted several times in the manuscript. As explicit consent for publication was given by the legal representatives we added this remark in the text.
- There are no international equivalents for the Dutch ID physician, so international comparison is not possible. Regarding ACE we presented some Dutch and international figures. As stated in the contribution, regarding PCE’s research in ID populations is scarce. Because of the limited purpose of the contribution namely an opinion article to be inspiring for international readers we hope this covers your remarks.
- Because of the opinionating and inspiring nature of this contribution it was decided not to follow this path.
Reviewer 3 Report
This manuscript stated some current issues regarding children and adolescents with 2 intellectual disabilities in the Netherlands. The topic is interesting, however, the structures of the article is not a scientific study. The whole study is only case basis.
Author Response
We thank the reviewer for the comments. Since our purpose was not to present a research article but to provide an opinionating and inspiring contribution we clarified this in the abstract and rewrote the article accordingly.
Round 2
Reviewer 2 Report
Review of the revised manuscript
A Dutch perspective on health related issues regarding children and adolescents with intellectual disabilities
I can see that the authors have rewritten most of the article. It definitely reads a lot better and for its intended purpose – the article is appropriate.
Here are some comments that I hope will help the authors improve their manuscript:
1. There are still some places where language editing is needed, although this issue was greatly improved compared with the first draft.
2. I would like the authors to provide a citation for the claim made at the start of chapter 3.1. As I have noted in my previous review, any factual statement needs to be supported by a reference. Now, regarding the practical aspects of the processes in the Netherlands and how ID physicians work – references would be welcome, but are not necessary since they discuss practical workflow specific to one country.
3. I would not call the brief description of Q’s case a “case study.” It is too brief for it to be a case study and it serves as an illustration of how ID physicians in the Netherlands add value to the care of people with ID.
4. Paragraphs still seem too long. Split them up into more manageable pieces. You want your article to be as easy to read as possible so that you can reach a wide readership.
5. Do you not need IRB approval for the case description you provided? I’m unfamiliar with regulations in The Netherlands regarding IRB approval. Please double-check.
I believe that after minor changes the article would be fit for publication.
I wish the authors the best of luck in their ongoing and future research.
Author Response
See below the responses to the reviewer
- There are still some places where language editing is needed, although this issue was greatly improved compared with the first draft.
Response: a second language editing was done.
- I would like the authors to provide a citation for the claim made at the start of chapter 3.1. As I have noted in my previous review, any factual statement needs to be supported by a reference. Now, regarding the practical aspects of the processes in the Netherlands and how ID physicians work – references would be welcome, but are not necessary since they discuss practical workflow specific to one country.
Response: we provided several references for the claim.
- I would not call the brief description of Q’s case a “case study.” It is too brief for it to be a case study and it serves as an illustration of how ID physicians in the Netherlands add value to the care of people with ID.
Response: we renamed this passage.
- Paragraphs still seem too long. Split them up into more manageable pieces. You want your article to be as easy to read as possible so that you can reach a wide readership.
Response: we did split paragraphs and adjusted the text order accordingly.
- Do you not need IRB approval for the case description you provided? I’m unfamiliar with regulations in The Netherlands regarding IRB approval. Please double-check.
Response: we double-checked this. No IRB approval was necessary.